# Outcomes of Treatment of Eyelids and Third Eyelid Tumours in Dogs Using High-Frequency Radiowave Surgery

**DOI:** 10.3390/ani13132105

**Published:** 2023-06-25

**Authors:** Luigi Navas, Cristina Di Palma, Maria Pia Pasolini, Chiara Montano, Mariaelena de Chiara, Francesco Lamagna, Valeria Uccello, Fabiana Micieli, Claudia Amalfitano, Orlando Paciello, Barbara Lamagna

**Affiliations:** 1Department of Veterinary Medicine and Animal Production, University of Naples Federico II, Via Federico Delpino 1, 80137 Naples, Italy; lnavas@unina.it (L.N.); cristina.dipalma@unina.it (C.D.P.); pasolini@unina.it (M.P.P.); mariaelena.dechiara@unina.it (M.d.C.); lamagna@unina.it (F.L.); dot.uccellovaleria@gmail.com (V.U.); fabiana.micieli@unina.it (F.M.); orlando.paciello@unina.it (O.P.); blamagna@unina.it (B.L.); 2Salus Vet, 80129 Napoli, Italy; claudia.amalfitano@gmail.com

**Keywords:** radiowave surgery, dog, eyelid tumour, third eyelid tumour, ophthalmic surgery

## Abstract

**Simple Summary:**

In canine eyelid surgery, intraoperative bleeding is a common feature that may obscure the surgical field view and lead to increased perioperative complications. High-frequency radiowave electrocautery has demonstrated several advantages in human medicine, including shorter surgery times and effective haemostasis. In this study, the records of 52 surgical excisions of canine eyelid and third eyelid tumours using high-frequency radiowave electrocautery were reviewed in order to evaluate the efficacy and benefits of this technique. Anamnesis, clinical signs, duration of surgery, success/recurrence rate, and complications were recorded. No intraoperative and postoperative complications were noted. The surgical incisions healed within 14 days in all the dogs, and no recurrences were reported in the 12 months after surgery. The use of high-frequency radiowave electrosurgery for the excision of eyelid and third eyelid tumours in dogs is effective and safe, without any intraoperative and postoperative complications in the 1-year follow-up period.

**Abstract:**

In human ophthalmology, the benefits of using high-frequency radiowave (HFR) electrocautery for surgical procedures were demonstrated and include effective haemostasis, shorter surgery times, and rapid recovery. In canine eyelid surgery, intraoperative bleeding is a common feature that may obscure the surgical field view and lead to the increased swelling of adjacent tissues, bruising, and pain. To evaluate the efficacy and benefits of HFR electrocautery in canine eyelid and third eyelid surgery, the medical records of 48 surgical excisions of eyelid tumours (involving up to one-third of the eyelid length) and 4 third eyelid excisions were reviewed. The information was collected including the breed, age, clinical signs, HFR power setting and mode of the surgical unit, electrode used for the surgery, intraoperative complications, histopathological diagnosis, and postoperative outcomes. Surgical techniques were performed using the Surgitron Dual 3.8 MHz Frequency RF device (Ellman International, Oceanside, NY, USA). Intraoperative bleeding was recorded as absent or very mild, and the surgical procedures were very fast. No complications occurred during the procedures. Healing within 10 days was observed in all the dogs. No tumour recurrences were recorded at the 12-month follow-up. HFR electrosurgery proved to be a safe, effective, and easy-to-perform technique for the removal of eyelid and third eyelid tumours in dogs.

## 1. Introduction

Intraoperative bleeding is a common feature during canine eyelid surgery due to the extensive vascularization of the periorbital region. This phenomenon may impair the vision of the surgical field and lead to postoperative swelling, bruising, and pain [1].

Compared to other species, eyelid tumours are very common in dogs [1], while third eyelid tumours, as well as other tumours of the conjunctiva, are uncommon in this species [2]. Therapies for canine eyelid and nictitating membrane tumours include surgical excision, cryosurgery, or carbon dioxide laser [1].

Surgical excision is effective, and its efficacy for larger tumours is higher than that of radiation or chemotherapy, although it may affect the normal structure and function of the eyelids [3].

Cryosurgery involves freezing the intracellular fluid, causing cell rupture in unwanted tissues. It is a simple and repeatable method, but it results in severe postoperative inflammation, discoloration, and the loss of normal tissue [4].

Carbon dioxide (CO_2_) laser produces an invisible beam of light of 10.600 nm, which is absorbed by the intracellular fluid, causing cellular vaporization and the ablation of neoplasms. This technique provides precise incision with good haemostasis, sanitization of the surgical field, and reduced tumour recurrence [5]. However, due to the risk of scattering and beam reflection, the use of personal protective equipment, including wavelength-specific eye protection, is required, and the apposition of the wound post laser ablation of meibomian gland adenomas is often performed due to frequent tissue loss in the eyelid margin [1].

In 1973, Dr. Irving Ellman developed a fully filtered waveform associated with a frequency of 3.8 MHz [5]. This frequency was optimal for producing a precise incision at the lowest temperatures, and Ellman patented the unit as the Dento-Surg Radiosurgical device [6]. The HFR unit works by converting electrical energy in a radiowave spectrum of 3.8 MHz; the normal alternating current is transformed into a direct current, which is then fed into a coil/capacitor or rectifier, to produce a radiowave signal. Radiowaves pass through a high-frequency waveform adapter that modifies the wave’s shape and magnitude; an amplifier increases the power level of the neo-produced waveforms [6]. Goldstein coined the term “radiosurgery” to distinguish the 3.8 MHz ultra-high-frequency radiowave device from electrosurgery (0.5–2.9 MHz) [6]. In 1978, Maness and colleagues showed that the HFR electrocautery limited uncontrolled thermal damage to tissues and achieved the smoothest cutting effects [7].

Since then, HFR electrocautery at 3.8–4 MHz frequencies have been used for the surgical treatment of various periodontal diseases, such as gingivectomy, gingivoplasty, crown lengthening, minimally invasive closed osteotomy, frenectomies, operculectomies, depigmentation, gingival curettage, periodontal flap surgery, mucogingival surgery, harvesting soft tissue grafts, and implantology [6]. HFR is also used in gynaecology, ophthalmology, plastic surgery, neurosurgery, arthrosurgery, otolaryngology, endoscopic spine surgery, and proctology [6].

In 1985, Stephen Bosniak introduced the radioelectrical device for oculoplastic surgery [5]. To date, high-frequency radiowave electrocautery has been used in human ophthalmology in various procedures, including punctoplasty, the correction of conjunctivochalasis, the removal of lymphangiectasia or conjunctival cysts/chemosis, and eyelid surgery [8,9,10,11,12,13].

In human eye surgery, HFR electrocautery was shown to be a precise, safe, and versatile technique that provides haemostasis and a shorter surgery time [9,10,11,12,13]. In addition, HFR technology minimizes heat dissipation and cellular alteration, minimizing postoperative discomfort and scar tissue formation, enhancing healing, and improving cosmetic outcomes [14].

In veterinary medicine, radiosurgery was described as a helpful tool in oral surgery [14] and in resection of the soft palate in brachycephalic dogs [15]. It was described as a useful option to reduce surgical time and postoperative swelling [16,17]. The use of radiosurgery has an essential role in the surgical treatment of many diseases in exotic animals; this device allows for the adequate control of haemostasis and a variety of electrodes, which allows for operation even in extremely small surgical fields [18,19]. HFR reduces anaesthesia and surgery time, which are particularly important factors in those species [6].

However, there are no published clinical studies to date concerning the treatment of eyelid and third eyelid tumours in dogs using HFR surgery.

Therefore, the aim of this retrospective study is to evaluate the efficacy and safety of HFR surgery for the treatment of eyelid and third eyelid tumours in dogs.

## 2. Materials and Methods

The medical records of dogs with eyelid or third eyelid tumours referred to the Teaching Veterinary Hospital of the University of Naples Federico II from January 2014 to January 2022 were examined.

This study includes records of dogs with eyelid masses (involving up to one-third of the eyelid length) or third eyelid tumours that underwent surgical resection using HFR electrosurgery and had a minimum follow-up of 12 months. Dogs that previously underwent eyelid surgery, were diagnosed with eyelid mast cell tumour, and had incomplete records were excluded. Data obtained from each clinical record include breed, age, clinical signs at presentation, HFR power setting and mode of the surgical unit, electrode used for the surgery, duration of surgery, intraoperative complications, histopathological diagnosis, and postoperative outcomes.

The same ophthalmologist performed a complete ophthalmic examination on both dog’s eyes, including neuro-ophthalmological examination and examination using STT-1 (Dina-HitexSpol, Bučovice, Czech Republic), slit-lamp biomicroscopy (Kowa SL-15, Kowa Company Ltd., Tokyo, Japan), direct ophthalmoscopy (Panoptic Ophthalmoscope, Welch Allyn), and measurement of intraocular pressure using a rebound tonometer (TonoVet, Icare Finland Oy, Vantaa, Finland).

The standard protocol included a general anaesthesia in sternal recumbency of all the patients.

Following anaesthetic premedication with intramuscular dexmedetomidine (0.005 mg/kg) and methadone (0.2 mg/kg), general anaesthesia was induced using effective propofol intravenously, and isoflurane or sevoflurane was used for maintenance of anaesthesia. The dose of both drugs was set to maintain an effective general anaesthesia. A bolus of fentanyl (0.002 mg/kg, IV) was administered in case of pain.

The periocular area was clipped and aseptically prepared using a 1:50 dilution of povidone-iodine solution. No local anaesthesia was administered.

An HFR surgical device (Ellman Surgitron FFPF EMC; output frequency: 3.8 MHz; continuously linear power setting) was used. The unit provides the following three different patterns of current flow or waveforms: “partially rectified” (indicated as “coag” on the dial), which produces an intermittent flow of waveforms; “fully rectified” (indicated as “cut & coag/rectified”), which provides waveforms in the same way as the cutting and coagulation effects, and is perceived as a minute pulsating waveform when used; and “fully rectified, filtered” (indicated as “filter cut”), which produces a continuous wave flow, resulting in a microsmooth cutting process with a 90% cutting effect and a 10% coagulation effect [6].

Eyelid tumours were excised using an empire microincision electrode (TEE305: length 3 cm, 45° angle) in cut mode. Eyelid masses involving up to one-third of the eyelid length were removed via a V-shaped or four-sided (house-shaped) incision at least 1 mm beyond the tumour margins, as recommended [1].

Bishop Harmon forceps were used to grasp the eyelid without inserting a lid plate underneath or placing a chalazion clamp over the mass. A V-shaped incision or the four-sided “house-shaped” incision was made using the empire microincision electrode starting from the skin surface, orthogonal to the palpebral plane. Power was set at 2.5 U and the incision was made through the skin, and then passed through the deeper tissues, increasing the power by half a unit or a whole unit as necessary. Care was taken to wait at least 10 s after each incision before approaching the surgical site with the electrode again.

The palpebral margin was sutured with a one-layer closure, using a figure-of-8 suture and 5-0 monofilament nylon (Ethilon^®^; Ethicon, Vienna, Austria). Then, the remaining lid was closed by employing a simple, interrupted pattern using 5-0 nonabsorbable monofilament sutures (Ethilon^®^; Ethicon, Vienna, Austria).

Dogs with third eyelid tumours underwent full third eyelid excision using the empire microincision electrode (TEE305: length 3 cm, 45° angle) in cut/coag mode.

An eyelid speculum was not applied. The third eyelid was elevated using Von Graefe forceps. Half the thickness of the front section of the base of the third eyelid was incised via HFR in “fully rectified” mode. Then, the third eyelid was pulled forward, and the posterior portion of the eyelid was incised. The defect was not sutured, allowing the conjunctiva to heal via second intention.

Intraoperative bleeding was classified by the surgeon as absent (no bleeding), mild (surgical field always visible), moderate (view of surgical field is problematic), or severe (surgical field can hardly be assessed).

In all cases, the passive electrode (ground plate) was positioned under the dog’s neck.

All excised tumours were sent for histopathologic examination at the Unit of Pathology of the Department of Veterinary Medicine and Animal Production at the University of Naples Federico II (Naples, Italy).

Postoperative treatment included the administration of 0.3% tobramycin topical drops (Tobral^®^, Alcon Italia S.p.A, Milan, Italy, TID), antibiotic therapy (cephazolin 20 mg/kg, IM, BID; or amoxicillin/clavulanic acid, 13.75 mg/kg, PO, BID), and analgesics (tramadol 2.5 mg/kg PO, BID). An Elizabethan collar was recommended for 2 weeks.

All dogs were re-examined ten days after surgery, then weekly for one month, and after two months. Lid sutures were removed at the first postoperative check-up. Additional information regarding recurrence or long-term complications was obtained during the examinations or via a phone interview with the owners or the referring veterinarian.

All numerical data were recorded using a computerized spreadsheet (Microsoft Excel ver.16.53 2, Microsoft Corp., Redmond, WA, USA). Results are expressed as mean ± standard deviation (SD) for parametric data, whereas non-parametric data are reported as median (IQR).

## 3. Results

A total of 52 records of dogs that underwent HFR electrosurgery were collected. The patient data are shown in Table 1 and Table 2.

The excision of the eyelid tumours was performed as described above in 48 dogs. Of all the dogs, 32 patients were males (61.5%), and 20 patients were females (38.5%). The median age was 7 years (range: 2 to 12 years). Mixed breed was the most common breed type (n = 13; 25%), followed by Yorkshire Terrier (n = 4; 7.7%), German Shepherd, Golden Retriever, Labrador Retriever (n = 3; 5.8%), Beagle, Boxer, Cocker Spaniel, English Setter, Pitbull, Poodle (n = 2; 3.8%), Dobermann, Saint Bernard, Maremma Shepherd, Dalmatian, Dogue de Bordeaux, Pinscher, Siberian Husky, Springer Spaniel, Breton, Rottweiler, Alaskan Malamute, Pekingese, Maltese, and Dachshund (n = 1; 1.9%).

The upper eyelid was involved in 26 cases (54.2%), whereas 22 cases had a lower eyelid disease (45.8%).

The eyelid tumours included meibomian gland adenoma (n = 24; 50%), papilloma (n = 10; 20.8%), meibomian gland epithelioma (n = 5; 10.4%), melanoma (n = 4; 8.3%), histiocytoma (n = 3; 6.2%), and meibomian gland adenocarcinoma (n = 2; 4.2%).

The excision of the third eyelid was performed as described above in four dogs. The median age was 7.5 years (range: 4 to 10 years). These cases included two male mixed-breed dogs, one female Labrador Retriever, and one male Golden Retriever. Two cases presented a pleomorphic carcinoma, and one presented a ductal carcinoma, while a histopathologic examination confirmed melanocytoma in one dog.

All surgical procedures were performed by the same experienced surgeon. The power was set at 2.5–4 U.

Intraoperative bleeding was recorded as absent (82% of the cases) or mild (18%), and the surgical procedures were very fast, with an average surgery duration time of 6.48 min ± 1.13 for an eyelid tumour excision and 6.8 min ± 0.84 for a third eyelid excision.

All patients presented mild conjunctival hyperaemia and slight swelling of the surgical site in the first postoperative day. At the ten-day postoperative examination, conjunctival hyperaemia and swelling were resolved in all cases (Figure 1, Figure 2 and Figure 3). All patients were considered healed 10 days after surgery, as shown in Figure 1, Figure 2 and Figure 3. No suture dehiscence was observed.

One dog with melanocytoma of the third eyelid developed postoperative depigmentation of the inferior lid margin after total third eyelid excision (Figure 4). However, two months after surgery, the depigmentation spontaneously resolved.

At the 12-month follow-up, the recurrence of neoplasm was not recorded in any of the cases, for which all the owners were satisfied with the resulting aesthetics. None of the dogs that underwent third eyelid excision developed a displacement of intraorbital fat or keratoconjunctivitis sicca.

## 4. Discussion

In this study, we investigated the effectiveness of high-frequency radiowave (HFR) surgery in treating tumours of the eyelids and nictitating membrane in dogs. This retrospective study confirms that the removal of eyelid or third eyelid tumours using HFR radiosurgery is safe and effective, with no significant complications and with a positive functional and cosmetic outcome.

Eyelid tumours are a common finding in dogs. Beagle, Siberian Husky, English Setter, Toy and Miniature Poodle, Labrador Retriever, and Golden Retriever were identified as breeds susceptible to developing these tumours [20]. All of these breeds are represented in our study, although epidemiological studies correlating the incidence of the pathology with the breed are still lacking [21].

Most eyelid tumours occur primarily in older dogs (over 10 years old) [20,22], and no gender predisposition was found [4]; however, in a study by Wang et al. (2019), female spayed dogs had a significantly higher prevalence of eyelid mass than female intact dogs. In our study, we show a greater susceptibility of male dogs to develop eyelid tumours. Moreover, the dogs’ median age was 7 years; this relatively young age may be related to the size (chosen for inclusion criteria) of the tumours, which were all small and histologically non-invasive. Viral papillomas and histiocytomas, in fact, occur more frequently in younger dogs [1,3,23], while squamous and reactive papillomas can occur in dogs of any age [24].

Up to 73% of eye neoplasms in canine species are benign [21,25]. In our study, we found that the most frequent eyelid tumours were neoplasms of the meibomian glands, as previously reported in reference [22]. In addition, the upper lid was affected slightly more often than the lower lid, as reported in reference [20]. Papillomas, meibomian gland tumours, melanomas, and histiocytomas were described as the most diagnosed eye tumours in this species [4]. In a study comprising 202 dogs, the most common eyelid tumours were sebaceous gland tumours (44%), papillomas (17.3%), and melanomas (20.8%) [26]. In another study comprising 200 dogs, a predominance of benign eyelid neoplasms was found, including 60% sebaceous adenomas, 17.6% melanomas, and 10.6% papillomas [20].

The most common neoplasia originating from the meibomian glands are adenomas, adenocarcinomas, and epitheliomas [18,27]. In our study, adenoma was the most diagnosed tumour of the meibomian glands. Labelle and Labelle (2013) showed no difference in the incidence of adenomas, epitheliomas, and carcinomas of the meibomian glands [28]; however, in a retrospective study by Muñoz-Duque et al. (2019) adenoma was the most frequently diagnosed neoplasm (22.8%), followed by epithelioma (20.0%), carcinoma (8.8%), and melanocytoma [29]. The differences between the studies may be related to the different geographical areas, number of included animals, ages, and breeds of the patients.

Among all eyelid tumours in dogs, melanocytic neoplasms have an incidence of 20%, and only 8% of them are melanomas [30]. Melanomas usually originate from eyelid skin and generally have a benign nature [31], but occasionally, some tumours can become more locally aggressive, involving the eyelid margin [30] and producing metastasis [32]. In the second case, surgery includes marginal excision and reconstruction [32]. In our study, all eyelid melanomas involved the eyelid margin; yet, there was no recurrence at the 12-month follow-up.

Given that they are frequently characterized as benign, small tumours do not require immediate removal; however, if the eyelid masses are rapidly growing, ulcerated, or produce corneal irritation, removal is indicated [21,33].

In domestic animals, surgical treatment options for commonly diagnosed eyelid neoplasms include surgical excision, cryotherapy, and laser surgery. Although these techniques have many advantages, they also present disadvantages that may lead to complications in the postoperative period [3,33]. Laser has excellent cosmetic results and a low recurrence rate. However, the equipment is expensive, and the use of protective devices by the surgeon is mandatory [33]. In our report, HFR surgery proved to be a satisfactory treatment option for the removal of lid and third eyelid tumours and is a less expensive treatment than CO_2_ laser that could be considered in canine ophthalmic surgery.

Radiosurgery generates high-frequency electricity at 3.8/4 MHz, which is delivered to the tissue through the ends of the electrodes [16,34]. The resistance of the tissue generates thermal energy that is passed to the intracellular and extracellular fluid, causing cellular coagulation and lysis [35].

This phenomenon, called “cellular volatilization”, leads to the coagulation and shrinkage of the target tissue; CO_2_ laser and HFR units generate thermal energy (65–75 °C), which vaporizes the intracellular water of the tissues [36].

HFR surgery (in incision–coagulation mode) was shown to minimize lateral heat damage in canine skin biopsies compared with monopolar electrosurgery and CO_2_ laser [37]. In addition, studies in different species have shown that radiosurgery causes significantly less collateral thermal damage in the skin and muscle than CO_2_ laser [18].

Bridenstine indicated a maximum thickness of heat-denatured collagen of 75 pm in biopsies performed using HFR during cosmetic surgeries; this value is comparable to that obtained using the carbon dioxide lasers [8]. A study comparing the use of CO_2_ laser, electrocautery, and HFR for the incision of human oviducts showed that HFR produced less damage to the surrounding tissue [38]. Furthermore, an analysis of the tissue margins of cone biopsy specimens obtained using a scalpel, CO_2_ laser, and a radiofrequency surgical unit in 40 patients showed similar results (less thermal and mechanical artifacts) between the HFR excision and scalpel removal [39].

In addition, research evaluating first-intention healing after CO_2_ laser surgery, 4.0 MHz radiowave radiosurgery, and scalpel incisions in ball pythons (*Python regius*), demonstrated that laser incisions had a significantly higher degree of dehiscence compared to HFR or scalpel incisions [36].

Nevertheless, it should be noted that the use of radiosurgery and laser devices on biological tissues produce a smoke plume that may contain pyrolysis products, which are known to be cytotoxic. This smoke, which may contain viable bacterial, viral, or patient DNA, should always be extracted from the surgical site using a filtered vacuum [18]. However, no other specific safety precautions are required for the HFR unit.

In human medicine, it was demonstrated that the use of HFR electrodevices in ophthalmic surgery produces simultaneous incision and coagulation with less surgical time, bleeding, and complications because of the reduced thermal injury of the surrounding tissues [9,10,11,18,34,35,40,41,42]. Reduced tissue damage leads to better scarring of the area and less discomfort during the postoperative period [43]. This may reduce the use of the Elizabethan collar, which is known to have a negative impact on animal health, nutrition, behaviour, and mental status [44].

During eyelid tumour excision, swelling is a common feature due to intense vascularization of the lids [43]. Intraoperative or postoperative bleeding was very mild in our cases, which allowed for the performance of accurate surgery with precise technique and resulted in smooth and regular eyelid margins.

Canine third eyelid tumours occur in older animals and may arise from the conjunctival or glandular tissues [25]. Melanomas, adenocarcinomas, squamous cell carcinomas, mastocytomas, papillomas, haemangiomas, haemangiosarcomas, angiokeratomas, and lymphosarcomas were reported in dogs [45]; adenocarcinomas were described as the most common primary tumours and are locally invasive in those animals [46]. Most vascular tumours of the third eyelid, as well as plasma cell tumours, are frequently surgically excised while sparing the third eyelid gland, while others, like melanomas, require third eyelid amputation [46].

In the treatment of third eyelid tumours, HFR offers the advantage of avoiding conjunctival suturing after total third eyelid excision without the herniation of intraorbital fat and with a good aesthetic result.

Postoperative complications after total third eyelid removal include decreased Schirmer tear test 1 and 2 values (STT-1 and 2), shortened tear film break-up times, and keratoconjunctival epithelial microinjury [47,48]. In a previous study, six dogs of different breeds and one cat were treated with CO_2_ laser for third eyelid total excision due to tumours involving either the glandular tissues or conjunctival surfaces of the nictitate membrane [49]. In this study, one patient developed a postoperative keratoconjunctivitis sicca, and the authors concluded that the pathophysiology needs further investigation to explain why some patients develop a decrease in tear production and others do not [50].

In our experience, we adopted all the general principles of radiowave surgery that are helpful to reduce additional thermal injury to the adjacent tissue (“lateral heat”) [6]. Choosing the optimal power settings and the right electrode are fundamental steps to avoid this damage. Furthermore, we used empire microincision electrodes, which produce a high concentration of energy with the least amount of thermal lateral damage. Additionally, we chose low power intensities (2.5–4 U) because it was shown that high power leads to excessive lateral heat due to sparking [16]. On the other hand, insufficient power also leads to lateral heat accumulation due to drag, which increases bleeding [1].

Monopolar electrodes require the use of a ground plate (passive electrode) to maximize radiosurgical efficiency and minimize lateral heat generation. This passive electrode should always be used and be positioned as close as possible to the surgical site. Therefore, we positioned the ground plate under the necks of all the patients.

Moving the electrode as quickly as possible is another important technique to achieve minimal contact time with tissues, minimizing the lateral heat produced [6]. We waited at least 10 s in each case before returning to the same surgical site to allow for tissue cooling, as recommended in the literature [6]. Finally, we used an HFR instrument set to pure cut mode for the eyelid excision, which resulted in less lateral tissue damage. Nevertheless, a third eyelid excision in one dog could have caused a minimal amount of lateral heat and reversible depigmentation of the inferior eyelid margin. The surgeon preferred to use the HFR instrument in cut/coagulation settings to achieve a better haemostatic effect.

In our study, the same surgeon performed the surgical procedure in all cases. In the future, it would be interesting to perform a multicentric study to evaluate whether the operator’s experience influences the postoperative outcome of HFR surgery. However, the use of HFR is no different from that of the scalpel, so it is easily learnable by surgeons. It should also be considered that this feature represents a further advantage offered by this device compared to the CO_2_ laser, that does not provide tactile sensation. The only difference may be related to the lack of tissue resistance, which is different from the one given by the scalpel and CO_2_ laser.

The twelve-month follow-up revealed no tumour recurrence in all of the treated dogs. The results are similar to those obtained with other techniques [50,51,52]. This study includes records of patients with small eyelid tumours and locally minimally invasive tumours (also meibomian adenocarcinomas) that are usually characterized by low recurrence rates [1]. Further studies evaluating the postoperative outcome after the HFR excision of more aggressive eyelid tumours are required.

No adjuvant therapy has been combined with the use of HFR; therefore, it would be interesting to investigate the combination of radiowave treatment and adjunctive local therapies in the future.

The research limitations are associated with the retrospective nature of this study and the relatively limited follow-up period. Further studies are needed to investigate the efficacy of HFR surgery in the removal of more aggressive and larger tumours and to compare the outcomes with other commonly used therapies.

## 5. Conclusions

In this study, high-frequency radiowave electrosurgery was found to be a safe, effective, and easy-to-perform technique for the excision of eyelid and third eyelid tumours in dogs. No functional alterations of the lids and no complications were noted in the postoperative period.

Further studies are needed in which HFR surgery is compared with other current treatments and in combination with other possible treatments.

## Figures and Tables

**Figure 1 animals-13-02105-f001:**
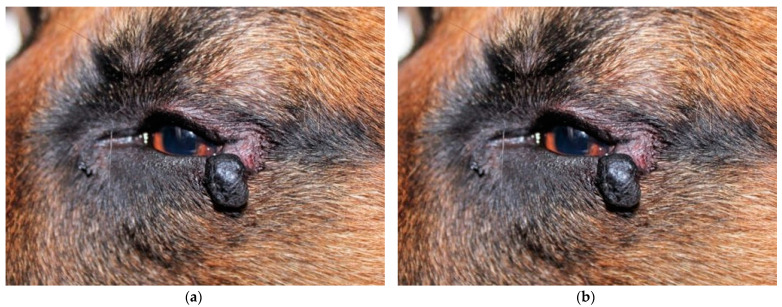
German Shepherd, male, 6 years old, with spindle cell melanoma (**a**) at presentation; (**b**) immediately after operation; (**c**) two weeks after surgery; (**d**) four weeks after surgery.

**Figure 2 animals-13-02105-f002:**
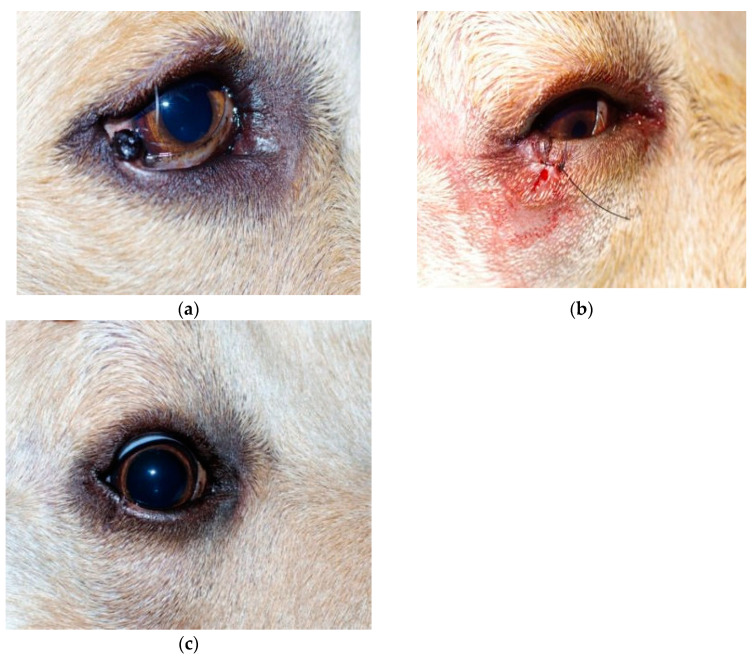
Labrador Retriever, male, 4 years old, with lid melanoma (**a**) at presentation; (**b**) immediately after operation; (**c**) four weeks after surgery.

**Figure 3 animals-13-02105-f003:**
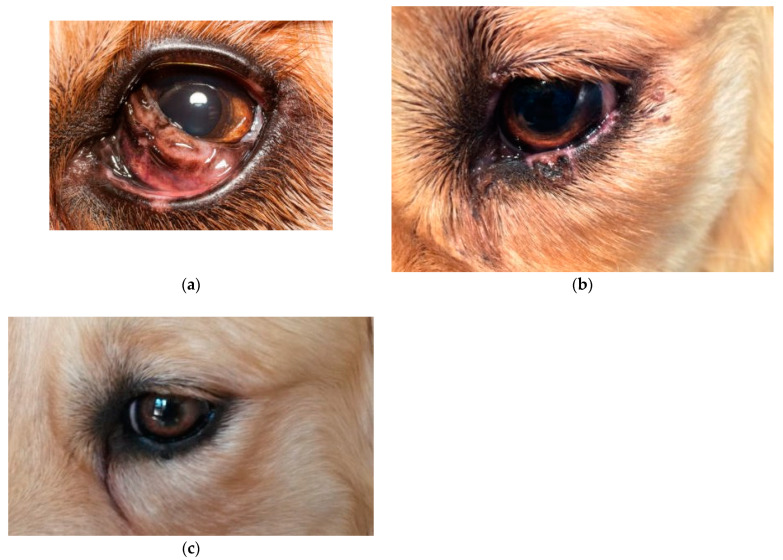
Golden Retriever, male, 4 years old, with third eyelid melanocytoma (**a**) at presentation; (**b**) two weeks after surgery, showing depigmentation of the inferior lid; (**c**) four weeks after surgery.

**Figure 4 animals-13-02105-f004:**
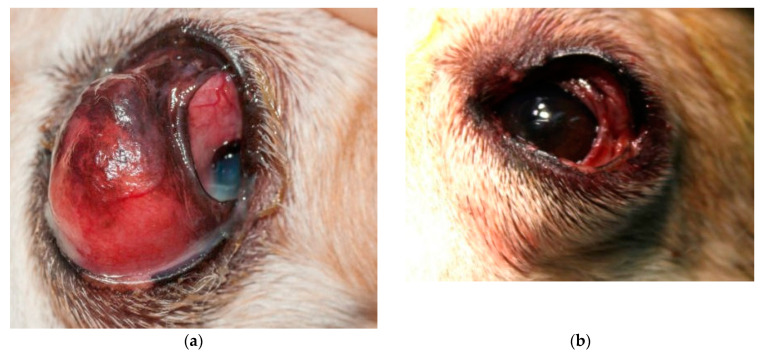
Mixed-breed dog, male, 10 years old, with adenocarcinoma of third eyelid (**a**) at presentation; (**b**) immediately post operation; (**c**) two weeks after surgery.

**Table 1 animals-13-02105-t001:** Breed, sex, age, type, localization of benign neoplasms, duration of surgery (minutes), and bleeding degree of dogs that underwent excision via HFR. M = male; F = female.

Breed	Sex	Age	Localization of the Tumor	Type of Tumor	Duration of Surgery	Bleeding Degree
Alaskan Malamute	F	7	Lower left eyelid	Meibomian gland adenoma	8	Absent
Beagle	M	7	Lower left eyelid	Meibomian gland epithelioma	7	Absent
Beagle	F	2	Upper right eyelid	Histiocytoma	7	Absent
Boxer	M	8	Upper right eyelid	Papilloma	7	Absent
Boxer	M	6	Upper right eyelid	Meibomian gland adenoma	5	Absent
Breton	F	8	Lower left eyelid	Papilloma	4	Absent
Cocker Spaniel	F	7	Upper right eyelid	Meibomian gland adenoma	6	Absent
Cocker Spaniel	M	8	Upper right eyelid	Meibomian gland adenoma	6	Absent
Dachshund	M	5	Upper left eyelid	Meibomian gland adenoma	6	Mild
Dalmatian	M	3	Upper left eyelid	Histiocytoma	6	Absent
Dobermann	M	7	Upper left eyelid	Papilloma	5	Absent
Dogue de Bordeaux	M	3	Lower right eyelid	Histiocytoma	7	Absent
English Setter	F	7	Upper right eyelid	Meibomian gland adenoma	5	Absent
English Setter	M	6	Upper right eyelid	Papilloma	6	Mild
German Shepherd	M	10	Lower left eyelid	Meibomian gland epithelioma	5	Absent
German Shepherd	M	6	Upper left eyelid	Meibomian gland adenoma	5	Absent
Golden Retriever	M	4	Left third eyelid	Melanocytoma	6	Absent
Golden Retriever	F	7	Upper right eyelid	Meibomian gland adenoma	5	Absent
Labrador Retriever	F	6	Lower right eyelid	Meibomian gland adenoma	7	Absent
Maltese	M	11	Lower left eyelid	Meibomian gland adenoma	6	Absent
Maremma Shepherd	F	6	Lower right eyelid	Meibomian gland adenoma	7	Absent
Mixed	M	9	Upper right eyelid	Meibomian gland epithelioma	8	Mild
Mixed	F	8	Upper right eyelid	Meibomian gland adenoma	6	Absent
Mixed	F	5	Upper right eyelid	Meibomian gland adenoma	8	Absent
Mixed	M	12	Upper left eyelid	Meibomian gland adenoma	7	Absent
Mixed	M	10	Lower right eyelid	Meibomian gland adenoma	5	Absent
Mixed	M	10	Upper right eyelid	Meibomian gland adenoma	7	Absent
Mixed	M	10	Upper left eyelid	Meibomian gland adenoma	7	Mild
Mixed	F	10	Upper right eyelid	Meibomian gland adenoma	6	Absent
Mixed	F	9	Lower right eyelid	Papilloma	7	Absent
Mixed	M	7	Lower right eyelid	Papilloma	9	Absent
Pekinese	M	8	Upper left eyelid	Meibomian gland adenoma	5	Absent
Pinscher	F	8	Lower right eyelid	Meibomian gland adenoma	5	Absent
Pitbull	M	7	Lower left eyelid	Meibomian gland adenoma	7	Mild
Pitbull	F	8	Upper right eyelid	Papilloma	8	Absent
Poodle	F	8	Lower left eyelid	Meibomian gland adenoma	6	Absent
Poodle	F	11	Lower left eyelid	Papilloma	8	Absent
Saint Bernard	M	5	Lower left eyelid	Meibomian gland adenoma	6	Absent
Siberian Husky	M	8	Upper right eyelid	Meibomian gland epithelioma	8	Absent
Springer Spaniel	M	6	Lower right eyelid	Papilloma	7	Absent
Yorkshire	F	10	Upper left eyelid	Meibomian gland adenoma	7	Absent
Yorkshire	M	8	Lower left eyelid	Papilloma	6	Absent
Yorkshire	M	7	Upper left eyelid	Meibomian gland epithelioma	9	Absent

**Table 2 animals-13-02105-t002:** Breed, sex, age, type, localization of malignant neoplasms, duration of surgery (minutes), and bleeding degree of dogs that underwent excision via HFR. M = male; F = female.

Breed	Sex	Age	Localization of the Tumour	Type of Tumour	Duration of Surgery	Bleeding Degree
German Shepherd	M	6	Lower left eyelid	Melanoma	7	Absent
Golden Retriever	M	6	Lower left eyelid	Melanoma	6	Absent
Labrador Retriever	F	7	Left third eyelid	Pleomorphic carcinoma	6	Mild
Labrador Retriever	M	4	Lower right eyelid	Melanoma	7	Absent
Mixed	M	10	Left third eyelid	Ductal carcinoma	8	Absent
Mixed	F	10	Upper right eyelid	Meibomian gland adenocarcinoma	6	Mild
Mixed	M	8	Right third eyelid	Pleomorphic carcinoma	7	Mild
Rottweiler	M	5	Lower right eyelid	Melanoma	7	Absent
Yorkshire	F	6	Upper left eyelid	Meibomian gland adenocarcinoma	6	Mild

## Data Availability

The data are available upon request from the submitting author.

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
