# Peer review of "Outcomes of Treatment of Eyelids and Third Eyelid Tumours in Dogs Using High-Frequency Radiowave Surgery"

_animals, 2023, doi:10.3390/ani13132105_

Round 1

Reviewer 1 Report

This study evaluated the efficacy and benefits of using high-frequency radiowave electrocautery for the excision of canine eyelid and third-eyelid tumours. Fifty-two canine patients underwent the procedure without any intraoperative or postoperative complications and all incisions healed within 14 days. No recurrences were reported in the 12-month follow-up period, suggesting that this technique is effective and safe with fewer complications.

The scientific methodology employed is correct and the information provided makes the study reproducible.

Graphical information is provided which aids reading and understanding.

The results are accurately expressed and data are presented in tables, including patient information.

An in-depth discussion is provided which offers insight and includes the authors' contributions based on their experience. This provides readers with useful information.

The conclusions are consistent with the results obtained.

With respect to the writing of the article, we believe it should be written by someone with English language proficiency. A revision of the text is recommended to ensure more accurate English expression.

Author Response

Thank you so much for your kind comment, we provided a complete English revision of the article.

Reviewer 2 Report

The manuscript “Outcomes of treatment of eyelids and third eyelid tumors in dogs using

high-frequency radiowave surgery” is concerned with an interesting surgical approach to eyelid and third eyelid tumor. The paper is straightforward and nicely written, there are only a few points concerning tumor classification that need to be clarified

Specifically:

-       The authors reported an elevated number of eyelid (dermal or palpebral rim) melanomas: based on the current pathological classification of melanocytic tumors, I assume that in authors’ caseload there was an unusually high number of cutaneous/rim malignant tumors. Surprisingly, this traditionally aggressive tumors had neither recurrence nor systemic diffusion in one year follow up. I have observed that the cited literature on melanocytic tumors is not recent, could the authors verify if the nomenclature applied to the present cases is consistent with the most recent indications?

-       Line 250: viral papillomas actually occur in young animals (but idiopathic papillomas do not, or not necessarily). However papillomas reported in the present caseload affected adult to old dogs, thus the hypothesis proposed applies, in the present cohort of cases, only to histiocytomas. The sentence should perhaps be re-formulated omitting papillomas

-       Which morphological type of third eyelid lacrimal gland carcinomas (e.g. adenoid cystic, pleomorphic, ….) have been observed?

-       Line 261-262: while meibomian adenomas are far the most common meibomian gland tumors in dogs, usually the number of meibomian gland carcinomas diagnosed is rather low.

The authors cited a chapter of Gelatt Veterinary Ophthalmology in which a table reported the percentage of eyelid tumors in two different large series. While adenocarcinomas were 15% of diagnosed tumors in the first study (1975), this percentage dropped to 2% in the more recent one (1986). Moreover, in these studies the diagnosis of Meibomian epithelioma was not considered. I suggest to refer to more recent publications for the relative frequency of meibomian gland tumor (e.g Veterinary Ocular Pathology a comparative review, but not necessarily this textbook)

-       321-322: this sentence should perhaps be slightly modified: most vascular neoplasia of the third eyelid, as well as plasma cell tumors, are frequently surgically excised sparing the third eyelid gland, while others (e.g. melanomas) are absolute indication for third eyelid amputation.

a few minor typing errors are scattered throughout the manuscript

Author Response

Thank you so much for your observations. We would like to answer to all your concerns:

  • The percentage of dogs presenting eye lid melanoma in our study was around 8.3%, which is even less that what we have found in the cited literature (Krehbiel et al., 1975; Robert et al., 1986). Melanomas involving the third eyelid and the upper eyelid are common in dogs. Usually, they are more locally aggressive than other eyelid tumours, however the prognosis is good after surgical removal (Gelatt & Plummer, 2022). As reported in the article, all treated tumours were not locally invasive. The use of the high-frequency radiowave surgery for the excision of tumours of the eyes showed low recurrence rate, that was 0% in our retrospective study, probably associated with the non-invasive histological characteristics of the treated tumours. Thank you for your comment, we are going to modify this section and the bibliography.
  • Thank you for your comment, we will change the sentence so as to clarify that viral papillomas are a common feature in young dogs, but squamous and reactive papillomas can occur in dogs of any age (Scott & Miller, 2020).
  • Thank you for your comment, we added the information about the morphological type in the revised version of the manuscript.
  • We check our data with the last edition of the Small Animal Ophthalmology (2022), that supports the information that we wrote in the paper. Also Labelle and Labelle (2013) showed no difference in the incidence of adenomas, epitheliomas, and carcinomas of the Meibomian glands. However, in a retrospective study by Muñoz-Duque et al. (2019) adenoma was the most frequent neoplasm diagnosed (22.8%), followed by epithelioma (20.0%), carcinoma (8.8%), and melanocytoma. Those difference may be related to the different geographic area, number of included animals, age and breed of the patients. We have modified the sentence with those observations. 
  • We have modified the sentence, thank you.

Reviewer 3 Report

This paper reports a retrospective study of dogs with eyelid and third eyelid (nictitating membrane) tumors, treated with high frequency radiowave (HFR) therapy for tumor removal.  The dogs were followed-up for at least 12 mos.  All procedures were performed by the same veterinary surgeon, presumably one of the authors.

 The surgical procedures carried out on the animals were described with sufficient detail to be reproduced by other practitioners.  The surgical results were favorable, notably with good hemostasis and rapid completion of the procedures.  There were no long-term complications noted and no tumor recurrence over the 12-month follow-up time.  It was interesting that in this study the median age of the subject dogs was relatively young (~7 years), compared to the findings of other studies of canine eyelid tumors.  The authors stated this was possibly due to the inclusion criteria used for the study animals.

 An interesting and useful discussion on the mechanism of HFR cutting and coagulation is presented in the Discussion.  The authors note correctly that HFR-tissue interaction is essentially via thermal tissue decomposition through vaporization of tissue water.  The important takeaway message from this study is that HFR is a cost-effective and safe method for removing tumors and can be used safely with delicate tissues such as the eye, because the deposition of the thermal energy is more spatially confined compared to other ablation methods, such as CO2 laser, contributing to an advantage in the use of HFR compared with other methods for eyelid tumor removal such as scalpel and laser ablation.  The authors also presented important safety precautions to be used with HFR, including smoke evacuation and careful setting of the operative power to minimize excessive thermal damage.  Overall, this is an interesting and useful report for a veterinary surgical application.

 One criticism is that the authors in the Methods section (lines 173-178) note the use of several statistical procedures for assessing significance of findings, but do not actually report any statistical results other than some percentages and median values.  Unless they can present some actual statistical tests on the data utilized in this study (perhaps they could compare the degree of hemostasis achieved in the procedures using HFR for tumor excision with the degree of hemostasis achieved with scalpel- or laser-based procedures), they may as well omit the statistical description.

 Minor Comments:

 1. Line 96: I confess I had no idea what a "palys" is.  The only online reference I found is that Palys is an exotic coral. Are the authors referring to this?  How/why is radiosurgery used on a coral?  I think the general readership of this journal would benefit from an explanation (or omit this statement).

2. Line 149: “approach” should be “approaching”

 3. Line 262: “melanoimas” probably should be “melanomas”

4. Line 297: “a research” should be “research”

4. Line 297: “a reaserch” should be “research”

The English writing and style is good; there are a few misspelled or misstated words, which I noted in the review.

Author Response

Thank you so much for your comments. We have corrected the errors that you have pointed out and modified the statistic section.

Round 2

Reviewer 1 Report

None

Author Response

Thank you!

Reviewer 3 Report

The authors have addressed very well the comments and critiques presented by the reviewers.  I have no further critiques to make on the manuscript and recommend it for publication.

Author Response

Thank you!